# Rotational symmetry breaking in superconducting nickelate Nd$_{0.8}$Sr$_{0.2}$NiO$_2$ films

Haoran Ji[1,16], Yi Liu [2,3,16], Yanan Li[1,4,16], Xiang Ding[5], Zheyuan Xie[1], Chengcheng Ji[1], Shichao Qi[1], Xiaoyue Gao[1,6], Minghui Xu[5], Peng Gao [1,6], Liang Qiao [5] ✉, Yi-feng Yang [7,8,9], Guang-Ming Zhang [10,11] & Jian Wang [1,12,13,14,15] ✉

The infinite-layer nickelates, isostructural to the high-$T_c$ cuprate super-conductors, have emerged as a promising platform to host unconventional superconductivity and stimulated growing interest in the condensed matter community. Despite considerable attention, the superconducting pairing symmetry of the nickelate superconductors, the fundamental characteristic of a superconducting state, is still under debate. Moreover, the strong electronic correlation in the nickelates may give rise to a rich phase diagram, where the underlying interplay between the superconductivity and other emerging quantum states with broken symmetry is awaiting exploration. Here, we study the angular dependence of the transport properties of the infinite-layer nickelate Nd$_{0.8}$Sr$_{0.2}$NiO$_2$ superconducting films with Corbino-disk configuration. The azimuthal angular dependence of the magnetoresistance ($R(\varphi)$) manifests the rotational symmetry breaking from isotropy to four-fold (C$_4$) anisotropy with increasing magnetic field, revealing a symmetry-breaking phase transition. Approaching the low-temperature and large-magnetic-field regime, an additional two-fold (C$_2$) symmetric component in the $R(\varphi)$ curves and an anomalous upturn of the temperature-dependent critical field are observed simultaneously, suggesting the emergence of an exotic electronic phase. Our work uncovers the evolution of the quantum states with different rotational symmetries in nickelate superconductors and provides deep insight into their global phase diagram.

The conventional superconductivity with transition temperature ($T_c$) lower than 40 K was successfully explained by the Bardeen-Cooper-Schrieffer (BCS) theory, in which the electrons with anti-parallel spins and time-reversed momenta form Cooper pairs, and the super-conducting order parameter is of isotropic $s$-wave symmetry[1,2]. However, the discovery of high-temperature superconductivity ($T_c > 40$ K) in cuprates is beyond the expectation of the BCS theory, and the superconducting order parameters of cuprates are believed to be of

nodal $d$-wave symmetry[3,4]. Thereafter, the mechanism of unconventional high-$T_c$ superconductivity has become one of the most important puzzles in physical sciences. Recently, the observation of superconductivity in infinite-layer nickelates with a maximal $T_c$ of 15 K in Nd$_{1-x}$Sr$_x$NiO$_2$ has motivated extensive researches in this emerging new superconducting family[5–9]. Mimicking the $d^9$ electronic config-uration and the layered structure including CuO$_2$ planes of the cup-rates, the isostructural infinite-layer nickelates are promising

candidates for high-$T_c$ unconventional superconductivity[5-9]. Discerning the similarities and the differences between the nickelates and the cuprates, especially in the symmetry of the superconducting order parameters, should be of great significance for understanding the mechanism of unconventional high-$T_c$ superconductivity.

Theoretical calculations have suggested that the nickelates are likely to give rise to a $d$-wave superconducting pairing, analogous to the cuprate superconductors. However, the consensus has not been reached and there are several proposals, including dominant $d_{x^2-y^2}$-wave[10,11], multi-band $d$-wave[12,13], and even a transition from $s$-wave to ($d+is$)-wave and then to $d$-wave depending on the doping level and the electrons hoping amplitude[14]. Experimentally, through the single-particle tunneling spectroscopy, different spectroscopic features showing $s$-wave, $d$-wave, and even a mixture of them are observed on different locations of the nickelate film surface, which complicates the determination of the pairing symmetry in the nickelates[15]. The London penetration depths of the nickelates family are also measured, and the results on La-based and Pr-based nickelate compounds support the existence of a $d$-wave component[16,17]. However, the Nd-based nickelate, $Nd_{0.8}Sr_{0.2}NiO_2$, exhibits more complex behaviors that may be captured by a predominantly isotropic nodeless pairing[16,17]. The pairing symmetry of the superconducting order parameter in the nickelate superconductors, the fundamental characteristic of the superconducting state, is still an open question, thus further explorations with diverse experimental techniques are highly desired.

In addition to the mystery of the superconducting pairing symmetry, the strong electronic correlation in nickelates is another element that makes the nickelate systems intriguing. The strong correlation is theoretically believed to play an important role in the nickelates systems[8,9,18,19] and the strong antiferromagnetic (AFM) exchange interaction between Ni spins has been experimentally detected[20]. Generally, the strongly correlated electronic systems are anticipated to host rich phase diagrams and multiple competing states including superconductivity, magnetic order, charge order, pair density wave (PDW), etc[21,22]. In the nickelate thin films, the charge order, a spatially periodic modulation of the electronic structure that breaks the translational symmetry, has been experimentally observed by the resonant inelastic X-ray scattering (RIXS)[23-25]. However, the charge order is only observable in the lower doping regime where the nickelates are non-superconducting. The interplay between the superconductivity and the charge order as well as other underlying symmetry-broken states is still awaiting explorations.

In this work, we report the polar and azimuthal angular-dependent transport behaviors of the superconducting state in infinite-layer nickelate $Nd_{0.8}Sr_{0.2}NiO_2$ films with the Corbino-disk configuration. The polar angular dependent magnetoresistance shows evident anisotropy, indicating the quasi-two-dimensional nature of the superconductivity. The azimuthal angular dependent magnetoresistance manifests a rotational symmetry breaking from isotropic to four-fold ($C_4$) rotational symmetric with increasing magnetic field. Strikingly, when approaching lower temperature and larger magnetic field regime, an additional two-fold ($C_2$) symmetric component emerges in the magnetoresistance, concomitant with an anomalous upturn of the temperature-dependent critical field. The observed successive rotational symmetry breakings in the magnetoresistance may uncover the subtle balance and the intriguing interplay between different competing orders in the $Nd_{0.8}Sr_{0.2}NiO_2$ thin films.

## Results

### Characterization of quasi-two-dimensional superconductivity

The perovskite precursor $Nd_{0.8}Sr_{0.2}NiO_3$ thin films are firstly deposited on the $SrTiO_3$ (001) substrates by pulsed laser deposition (PLD). The apical oxygen is then removed by the soft-chemistry topotactic reduction method using $CaH_2$ power. Through this procedure, the

nickelate thin films undergo a topotactic transition from the perovskite phase to the infinite-layer phase, and thus the superconducting $Nd_{0.8}Sr_{0.2}NiO_2$ thin films are obtained[5]. Figure 1a presents the schematic crystal structure of $Nd_{0.8}Sr_{0.2}NiO_2$. In agreement with the previous reports[5], the temperature-dependence of the resistance $R(T)$ exhibits metallic behavior from room temperature to low temperature followed by a superconducting transition beginning at $T_c^{onset}$ of 14.8 K (Fig. 1b). Here, $T_c^{onset}$ is determined at the point where $R(T)$ deviates from the extrapolation of the normal state resistance ($R_N$). Note that the $R(T)$ curve shows a considerably broad superconducting transition with a smooth tail, which can be described by the Berezinskii-Kosterlitz-Thouless (BKT) transition in two-dimensional (2D) superconductors[26-29]. As shown in the inset of Fig. 1b, the $R(T)$ curve under 0 T can be reproduced by the BKT transition using the Halperin-Nelson equation[30], $R = R_0 \exp\left[-2b\left(\frac{T_c'-T_{BKT}}{T-T_{BKT}}\right)^{1/2}\right]$ ($R_0$ and $b$ are material-dependent parameters, and $T_c'$ is the superconducting critical temperature), yielding the BKT transition temperature $T_{BKT}$ of 8.5 K. An apparent difference is also noted between the $R(T)$ curves under in-plane and out-of-plane magnetic fields (inset of Fig. 1b), implying the anisotropy of the superconductivity.

To obtain more insight into the anisotropic superconductivity in $Nd_{0.8}Sr_{0.2}NiO_2$ thin films, the critical magnetic field and magnetoresistance under different magnetic field orientations are measured. Here, the Corbino-disk configuration is used to eliminate the influence of the current flow in angular-dependent magnetoresistance measurements[31], which cannot be completely avoided in standard four-probe measurements[32,33]. The schematic image and the optical photo of a Corbino-disk device are shown in Fig. 1c. To start with, the temperature-dependence of the critical field $B_c$ is measured under the magnetic field applied along the $c$-axis (denoted as $\perp$), the $a/b$-axis ($\parallel$, 0°), and the $ab$ diagonal direction ($\parallel$, 45°). Here, $B_c$ is defined as the magnetic field required to reach 50% of the normal state resistance ($R_N = 98.8\ \Omega$), and the $B_c(T)$ curves are collected in Fig. 1d. The $T$-linear dependence of $B_{c\perp}(T)$, and the $(T_c-T)^{1/2}$-dependence of $B_{c\parallel,0°}$ and $B_{c\parallel,45°}$ near $T_c$ show agreement with the phenomenological 2D Ginzburg-Landau (G-L) formula[34]:

$$B_{c\perp}(T) = \frac{\phi_0}{2\pi\xi_{G-L}^2(0)}\left(1-\frac{T}{T_c}\right) \tag{1}$$

$$B_{c\parallel}(T) = \frac{\sqrt{12}\phi_0}{2\pi\xi_{G-L}(0)d_{sc}}\left(1-\frac{T}{T_c}\right)^{\frac{1}{2}} \tag{2}$$

where $\phi_0$ is the flux quantum, $\xi_{G-L}(0)$ is the zero-temperature G-L coherence length and $d_{sc}$ is the superconducting thickness. The consistency with the 2D G-L formula near $T_c$ indicates the 2D nature of the superconductivity in the $Nd_{0.8}Sr_{0.2}NiO_2$ thin films. To further study the dimensionality of the superconductivity, we measure the polar angular dependence of the critical magnetic field $B_c(\theta)$ for $Nd_{0.8}Sr_{0.2}NiO_2$ thin film at $T = 6$ K. Here, $\theta$ represents the angle between the magnetic field and the $c$-axis of the $Nd_{0.8}Sr_{0.2}NiO_2$. As shown in Fig. 1e, the $B_c(\theta)$ curve exhibits a prominent angular dependence of the external magnetic field, and a cusp-like peak is clearly resolved around $\theta = 90°$ ($B \perp c$-axis). The peak in the $B_c(\theta)$ around 90° can be well reproduced by the 2D Tinkham model and cannot be captured by the 3D anisotropic mass model, which qualitatively demonstrates the behavior of 2D superconductivity[35] (inset of Fig. 1e).

### Polar angular dependence of the magnetoresistance $R(\theta)$

To obtain a more comprehensive depiction of the anisotropy, the polar angular dependence of the magnetoresistance $R(\theta)$ at various

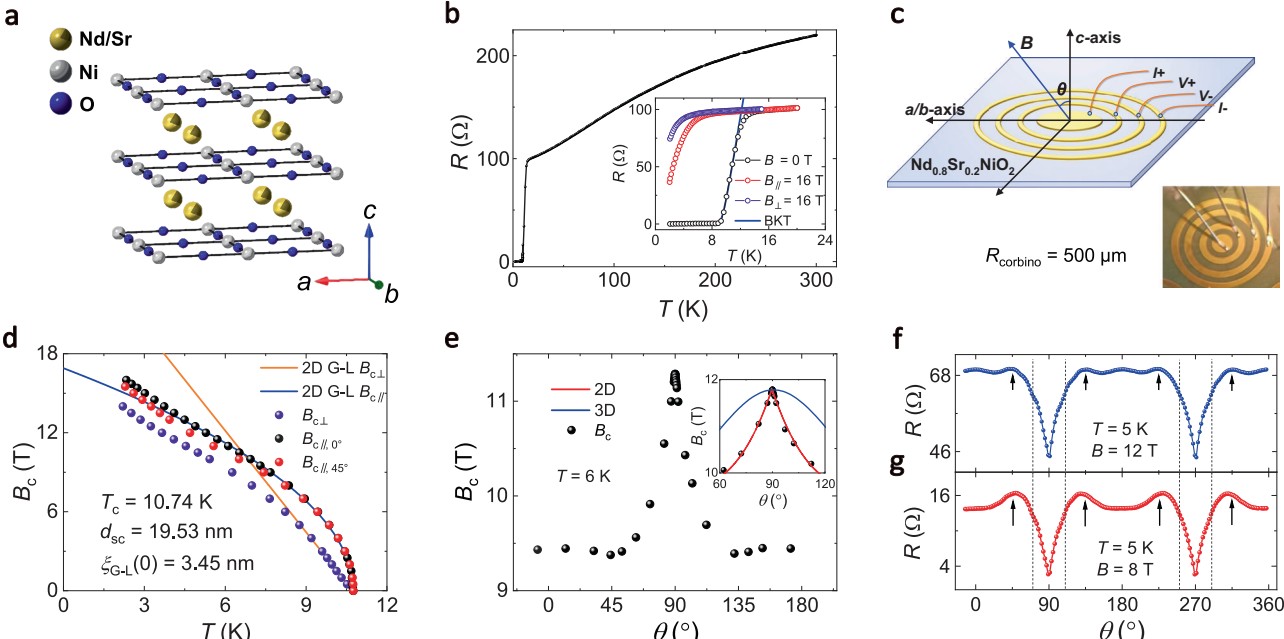

**Fig. 1 | Structure and the quasi-two-dimensional superconductivity in Nd$_{0.8}$Sr$_{0.2}$NiO$_2$. a** Crystal structure of the infinite-layer nickelate Nd$_{0.8}$Sr$_{0.2}$NiO$_2$. **b** Temperature dependence of the resistance $R(T)$ at zero magnetic field from 2 K to 300 K. The inset shows the $R(T)$ curves below 20 K at 0 T (black circles), $B_{\parallel} = 16$ T (red circles), and $B_{\perp} = 16$ T (purple circles). Here, the $B_{\parallel}$ is applied along the $a/b$-axis and $B_{\perp}$ is along the $c$-axis. The blue solid line represents the BKT transition fitting using the Halperin-Nelson equation. **c** Schematic image and optical photo (inset) of the Corbino-disk configuration for polar ($\theta$) angular dependent magnetoresistance $R(\theta)$ measurements on the Nd$_{0.8}$Sr$_{0.2}$NiO$_2$ thin film. Here, $\theta$ represents the angle between the magnetic field and the $c$-axis of the Nd$_{0.8}$Sr$_{0.2}$NiO$_2$. **d** Temperature dependence of the critical magnetic field $B_c(T)$ for

the magnetic fields along the $c$-axis (denoted as $\perp$), the $a/b$-axis ($\parallel$, 0°), and the $ab$ diagonal direction ($\parallel$, 45°). Here, the $B_c$ is defined as the magnetic field corresponding to 50% normal state resistance. The blue and the orange solid lines are the 2D G-L fittings of the $B_c(T)$ data near $T_c$. **e** Polar angular dependence of the critical magnetic field $B_c(\theta)$ at $T = 6$ K. The inset shows a close-up of the $B_c(\theta)$ around $\theta = 90°$. The red solid line and the blue solid line are the fittings with the 2D Tinkham model $(B_c(\theta)\sin(\theta)/B_{c\parallel})^2 + |B_c(\theta)\cos(\theta)/B_{c\perp}| = 1$ and the 3D anisotropic mass model $B_c(\theta) = B_{c\parallel}/(\sin^2(\theta) + \gamma^2\cos^2(\theta))^{1/2}$ with $\gamma = B_{c\parallel}/B_{c\perp}$, respectively. **f, g** Representative polar angular dependence of the magnetoresistance $R(\theta)$ at temperature $T = 5$ K under $B = 12$ T (**f**) and $B = 8$ T (**g**).

temperatures and magnetic fields are measured, and two representative $R(\theta)$ curves at 5 K under 12 T and 8 T are shown in Fig. 1f and g, respectively. The most notable features are two sharp dips at 90° and 270°, corresponding to $B \perp c$-axis. The two sharp dips correspond to the cusp-like peak in the $B_c(\theta)$ curve, resulting from the quasi-2D anisotropy. With varying temperatures and magnetic fields, the two sharp dips are observed in all $R(\theta)$ curves measured in the superconducting region (Supplementary Fig. 11), further confirming the quasi-2D nature of the superconducting Nd$_{0.8}$Sr$_{0.2}$NiO$_2$ thin films and suggesting that the layered superconducting NiO$_2$ planes should mainly account for the superconducting properties in our transport measurements. Additionally, small humps at approximately 90° ± 45° and 270° ± 45° are observed under 8 T and 12 T (marked by black arrows in Fig. 1f and g, respectively), while four more subtle kinks at 90° ± 20° and 270° ± 20° can be seen under 12 T (marked by black dashed line Fig. 1g). Considering the crystal structure of the Nd$_{0.8}$Sr$_{0.2}$NiO$_2$, the humps and kinks with relatively small variations may originate from the spin-dependent electron scattering with the magnetic moment of the rare-earth Nd$^{3+}$. Further investigations are still desired to prove this scenario.

**Azimuthal angular dependence of the magnetoresistance $R(\varphi)$**

It is noteworthy that in Fig. 1d, the $B_{c\parallel,0°}(T)$ and $B_{c\parallel,45°}(T)$ curves overlap with each other below ~4 T (above 9 K), but split with increasing magnetic field (decreasing temperature), suggesting the existence of in-plane anisotropy of the superconductivity of the Nd$_{0.8}$Sr$_{0.2}$NiO$_2$ thin films. Thus, we performed the azimuthal angular dependence of the magnetoresistance $R(\varphi)$ to reveal the in-plane anisotropy of the Nd$_{0.8}$Sr$_{0.2}$NiO$_2$ thin films using the Corbino-disk configuration, as

schematically shown in Fig. 2a. Here, $\varphi$ represents the angle between the magnetic field and the $a/b$-axis of the Nd$_{0.8}$Sr$_{0.2}$NiO$_2$. The Corbino-disk configuration guarantees that the electric current flows radially from the center to the outermost electrode, which well excludes the undesired influence of the current flow and ensures that the measured anisotropy is the intrinsic properties of the Nd$_{0.8}$Sr$_{0.2}$NiO$_2$ thin films. The representative set of $R(\varphi)$ at different temperatures under 8 T in polar and rectangular plots are shown in Fig. 2b, c, respectively (more sets of $R(\varphi)$ at different temperatures and magnetic fields can be found in the Supplementary Information). Remarkably, the $R(\varphi)$ curves exhibit obvious four-fold (C$_4$) rotational symmetry in both polar and rectangular plots. The C$_4$ symmetry of the $R(\varphi)$ curves shows minima at 0°, 90°, 180°, and 270° (along the $a/b$-axis) and maxima at 45°, 135°, 225°, and 315° (45° to the $a/b$-axis) from 8 K to 11 K, and becomes indistinguishable when the temperature is increased to 14 K (the top panel in Fig. 2c).

To confirm the correspondence between the minima of the C$_4$ $R(\varphi)$ and the $a/b$-axis of the Nd$_{0.8}$Sr$_{0.2}$NiO$_2$ thin films, control experiments have been carefully conducted. Specifically, the Nd$_{0.8}$Sr$_{0.2}$NiO$_2$ Corbino-disk device is remounted and remeasured after rotating a finite in-plane degree $\Delta\varphi$. Through the comparison between the initial results $R(\varphi)$ and the remeasured results after rotation $R(\varphi + \Delta\varphi)$, the minima (maxima) of the C$_4$ symmetry are fixed with the $a/b$-axis (45° to the $a/b$-axis), verifying the C$_4$ symmetry of the $R(\varphi)$ is an intrinsic property of the Nd$_{0.8}$Sr$_{0.2}$NiO$_2$ thin films (Supplementary Fig. 12). Note that the $R(\varphi)$ curve at 14 K is already larger than the normal state resistance $R_N$ of 98.8 Ω, and is almost isotropic within the experimental resolution. Thus, the observed C$_4$ symmetry below 11 K should be related to the superconducting characteristics of the Nd$_{0.8}$Sr$_{0.2}$NiO$_2$

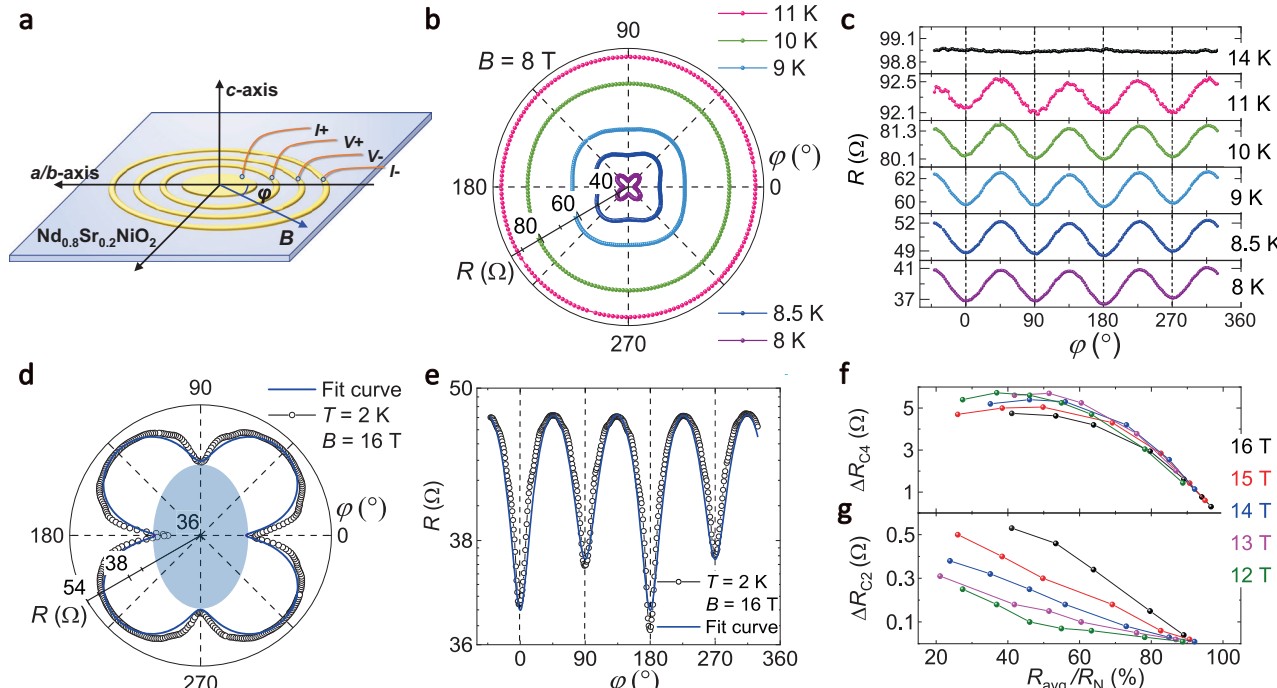

**Fig. 2 | Azimuthal ($\varphi$) angular dependence of the magnetoresistance in Nd$_{0.8}$Sr$_{0.2}$NiO$_2$.** **a** Schematic of the Corbino-disk device for azimuthal ($\varphi$) angular dependent magnetoresistance measurements. Here, $\varphi$ represents the angle between the magnetic field and the $a/b$-axis of the Nd$_{0.8}$Sr$_{0.2}$NiO$_2$. **b, c** Azimuthal angle dependence of the magnetoresistance $R(\varphi)$ at different temperatures under $B = 8$ T in the polar plot (**b**) and rectangular plot (**c**). **d, e** Azimuthal angle dependence of the magnetoresistance $R(\varphi)$ at $T = 2$ K under $B = 16$ T in the polar plot (**d**) and rectangular plot (**e**). Here, the logarithmic scale is used on the resistance-axis to

specifically demonstrate the C$_2$ symmetric feature. The blue solid lines are fits with the trigonometric function: $R = R_{avg} + \Delta R_{C4} \times \sin(4\varphi) + \Delta R_{C2} \times \sin(2\varphi)$, where $R_{avg}$ is the averaged magnetoresistance and $\Delta R_{C4}$ and $\Delta R_{C2}$ are the C$_4$ and C$_2$ components, respectively. The light blue area in (**d**) is a guide to the eye, representing the C$_2$ anisotropy. **f, g** Four-fold components $\Delta R_{C4}$ (**f**) and two-fold components $\Delta R_{C2}$ (**g**) versus the ratio between the averaged magnetoresistance and the normal state resistance ($R_{avg}/R_N$) under different magnetic fields. Here, the values of the C$_2$ and C$_4$ components are extracted by the trigonometric function fitting.

thin films, evidenced by that the C$_4$ anisotropy disappears as the superconductivity is suppressed at higher temperatures. Moreover, considering that the quasi-2D nature of the superconductivity in the Nd$_{0.8}$Sr$_{0.2}$NiO$_2$ and the large magnetoresistance amplitudes of the C$_4$ anisotropy ($\Delta R_{C4}/R$) (approximately 10% under 8 K and 8 T in Fig. 2c, around 20% under 5.5 K and 12 T in the Supplementary Fig. 14), the C$_4$ symmetry is not likely owing to the magnetic moment of the Nd$^{3+}$ between the NiO$_2$ planes, but should be ascribed to the superconductivity in the NiO$_2$ planes. The possible role of the Nd$^{3+}$ magnetic moment in our $R(\varphi)$ measurements is further discussed in the Supplementary Information. The C$_4$ anisotropy has ever been observed in the normal state of the cuprates with a magnitude of merely 0.05%, which is attributed to the magnetic order[36] and different from our observations. The in-plane critical field of a $d$-wave superconductor is theoretically predicted to exhibit a C$_4$ symmetric anisotropy owing to the $d$-wave pairing symmetry[37]. The C$_4$ anisotropic critical field as well as the C$_4$ anisotropic magnetoresistance has been experimentally used to determine the $d$-wave superconductivity in cuprate superconductors[38] and heavy fermion superconductors[39], etc. Therefore, the C$_4$ symmetry of our $R(\varphi)$ curves is supposed to imply the C$_4$ symmetric critical field of the predominant $d$-wave pairing in the Nd$_{0.8}$Sr$_{0.2}$NiO$_2$ thin films. The deduced $d$-wave pairing in the Nd$_{0.8}$Sr$_{0.2}$NiO$_2$ thin films cannot be definitely determined by our transport measurement results solely and requires further experimental investigations (e.g., phases-sensitive measurements).

Remarkably, with further increasing magnetic field, additional two-fold (C$_2$) symmetric signals are observed as small modulations superimposed on the primary C$_4$ symmetry in the $R(\varphi)$ curves. The representative $R(\varphi)$ curve under 2 K and 16 T in the polar and rectangular plots are shown in Fig. 2d and e, respectively. In addition to the predominant C$_4$ symmetric $R(\varphi)$, an additional C$_2$ signal can be clearly

discerned by $R(\varphi = 0°$ and 180°) being smaller than $R(\varphi = 90°$ and 270°), indicating the rotational symmetry breaking between $a$-axis and $b$-axis. In the following, elaborated experiments and analysis are discussed to exclude the possible extrinsic origin of the C$_2$ anisotropy. First, a nearly perfect C$_2$ symmetric feature is confirmed through a two-axis stage rotator, which can basically eliminate the misalignment of the applied magnetic fields (Supplementary Fig. 24). Second, through the aforementioned remounted measurements after an in-plane rotation of $\Delta\varphi$, the C$_2$ symmetry is confirmed to be invariant with respect to the sample mounting, since $R(\varphi + \Delta\varphi)$ can nicely overlap with $R(\varphi)$ after slight shifts (Supplementary Fig. 12). Third, the Corbino-disk configuration excludes the anisotropic vortex motion due to the uni-directional current flow, which has been reported in the previous works using the standard four-probe or Hall-bar electrode[32,33]. Fourth, the C$_2$ features superimposed on the C$_4$ symmetric $R(\varphi)$ can be consistently observed in many other samples in the large magnetic field, demonstrating the strong reproducibility of the C$_2$ and C$_4$ anisotropy (Supplementary Fig. 25). Fifth, the amplitudes of the C$_2$ anisotropy in our samples (>1 Ω) are orders of magnitude larger than the resistance resolution of our measurement equipment (20 mΩ with the excitation current of 1 μA).

To quantitatively study the evolution of the C$_2$ and C$_4$ anisotropy of the $R(\varphi)$, the C$_2$ components ($\Delta R_{C2}$) and C$_4$ components ($\Delta R_{C4}$) of each $R(\varphi)$ curve at different temperatures and magnetic fields are extracted through trigonometric function fitting (Supplementary Fig. 15). Among them, the fitting curve of the $R(\varphi)$ under 2 K and 16 T is shown in Fig. 2d, e. Here, the ratio between the average resistance of the $R(\varphi)$ and the normal state resistance ($R_{avg}/R_N$) is used as an independent variable for an intuitive comparison. Figure 2f shows $\Delta R_{C4}$ as a function of $R_{avg}/R_N$ under different magnetic fields, which exhibit similar parabolic behaviors with maxima approximately around 50%

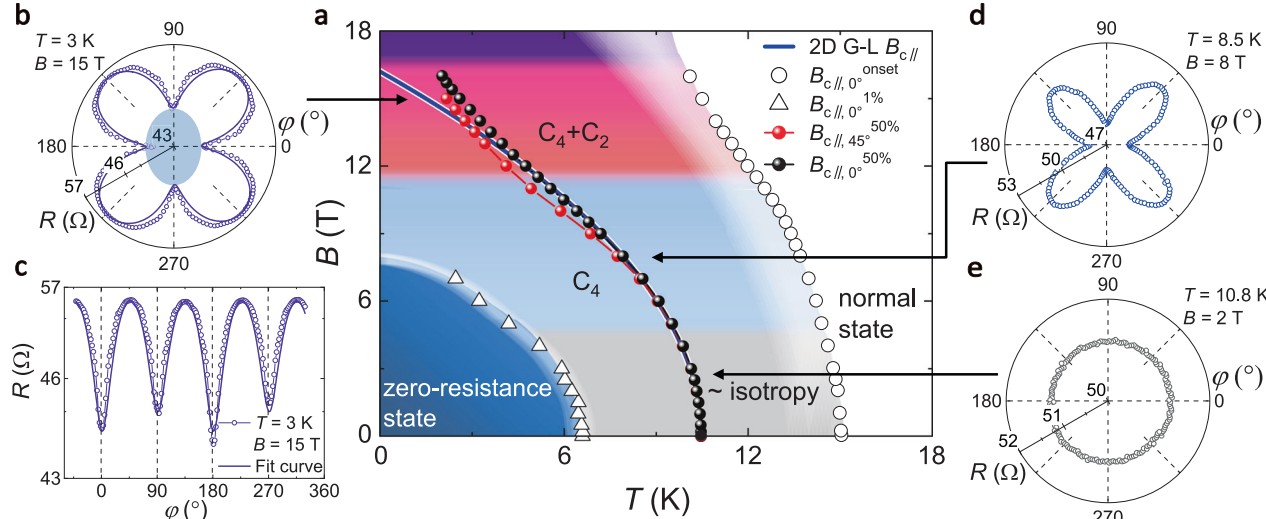

**Fig. 3 | B versus T phase diagram for the Nd$_{0.8}$Sr$_{0.2}$NiO$_2$. a** $B$ versus $T$ phase diagram for in-plane magnetic fields. The white region above $B_{c\parallel,0°}^{onset}(T)$ (open circles) represents the normal state, and the dark blue region below $B_{c\parallel,0°}^{1\%}(T)$ (open triangles) denotes the zero-resistance state (defined by $R < 1\%$ of $R_N$). Between $B_{c\parallel,0°}^{onset}$ and $B_{c\parallel,0°}^{1\%}(T)$ is the superconducting transition region, which is separated into ~isotropy (gray area), C$_4$ (blue area), and C$_4$ + C$_2$ (pink area) regions from small-magnetic-field to large-magnetic-field regime, depending on whether the measured $R(\varphi)$ curves exhibit nearly isotropy, C$_4$ or C$_4$ + C$_2$ rotational symmetry. $B_{c\parallel,0°}^{50\%}(T)$ (black dots) and $B_{c\parallel,45°}^{50\%}(T)$ (red dots) represent the critical field along the $a/b$-axis (0°) and the $ab$ diagonal direction (45°) determined by the 50%

$R_N$ criterion, respectively. The blue solid line is the 2D G-L fit of the $B_{c\parallel,0°}^{50\%}(T)$ data near $T_c$. The $B_{c\parallel,0°}^{50\%}(T)$ and $B_{c\parallel,45°}^{50\%}(T)$ data are from sample S2, which show a more prominent upturn in the low-temperature regime. **b**–**e** The representative $R(\varphi)$ curves at 3 K and 15 T in the polar plot (**b**) and rectangular plot (**c**), 8.5 K and 8 T in the polar plot (**d**) and 10.8 K and 2 T in the polar plot (**e**), manifesting different rotational anisotropies. $\varphi$ represents the azimuthal angle between the magnetic field and the $a/b$-axis of the Nd$_{0.8}$Sr$_{0.2}$NiO$_2$. The light blue area in **b** is a guide to the eye, representing the C$_2$ anisotropy. The black arrows in this figure approximately mark the temperature and magnetic field positions in the phase diagram where the $R(\varphi)$ curves are measured.

$R_N$. Differently, the $\Delta R_{C2}$ are monotonically decreasing with increasing $R_{avg}/R_N$ as shown in Fig. 2g. With increasing magnetic field, the $\Delta R_{C4}$ show subtle decreasing tendency, while the $\Delta R_{C2}$ monotonically increase, exhibiting a magnetic field-mediated competition between the $\Delta R_{C4}$ and the $\Delta R_{C2}$. The parabolic $R_{avg}/R_N$-dependence of the $\Delta R_{C4}$ can be understood as the resistance anisotropy due to the super-conductivity anisotropy becomes smaller when approaching the superconducting zero-resistance state or the normal state. As the sample is approaching the normal state and losing the super-conductivity ($R$ getting close to $R_N$ as well as $T$ getting close to $T_c^{onset}$), the amplitudes of the C$_4$ anisotropy are diminishing, which unam-biguously shows that the C$_4$ anisotropy is associated with the super-conducting state. This coincidence between the C$_4$ anisotropy and superconducting transition is verified for a range of $T_c$, which varies with applied magnetic fields and among different samples (Supple-mentary Figs. 22 and 23). However, the monotonic $R_{avg}/R_N$-depen-dence of the $\Delta R_{C2}$ cannot be explained by such a scenario, suggesting a different origin of the C$_2$ symmetry. Generally, the observation of spontaneous rotational symmetry breaking in $R(\varphi)$ curves would indicate the existence of nematicity[31,40]. However, in our measure-ments, the C$_2$ component has a relatively small weight in the aniso-tropy of the $R(\varphi)$ compared with the C$_4$ symmetry (<12%), inconsistent with previous results of nematic superconductivity with a primary C$_2$ feature[31]. To understand the origin of the C$_2$ anisotropy, we recall the RIXS-detected charge order in the nickelates that is along the Ni-O bond direction and exhibits a competitive relationship with the superconductivity[23–25]. Considering the Ni-O bond direction of our C$_2$ feature and the magnetic field-mediated competition between the $\Delta R_{C4}$ and the $\Delta R_{C2}$, our observation of the C$_2$ anisotropy might result from the charge order in Nd$_{0.8}$Sr$_{0.2}$NiO$_2$ thin films. The magnetic field suppresses the superconductivity and may alter the competition between the anisotropic superconductivity with C$_4$ symmetry and the charge order fluctuations with C$_2$ symmetry in our Nd$_{0.8}$Sr$_{0.2}$NiO$_2$, leading to the monotonically magnetic field-dependent decrease of the $\Delta R_{C4}$ (Fig. 2f) and increase of the $\Delta R_{C2}$ in our observations (Fig. 2g).

As previously reported, the Sr doping dramatically lowers the onset temperature of the charge order in the nickelates[23], which may explain the occurrence of the charge order in our Nd$_{0.8}$Sr$_{0.2}$NiO$_2$ only at low temperatures. Although the previous RIXS results show that under zero magnetic field the charge order is absent at a doping level of 0.2[23–25], the strong magnetic fields used in our measurements may suppress the superconductivity and raise the charge order fluctua-tions. In addition, since the C$_2$ feature breaks the rotational symmetry, our observation favors a stripe-like charge order in the nickelates, which is supported by theoretical proposals[18,19], while further investi-gations are still needed.

## Magnetic field versus temperature phase diagram
Based on the $B_c(T)$ curves and the anisotropic behaviors of the $R(\varphi)$, we construct the global phase diagram (Fig. 3a) to develop a compre-hensive understanding of the superconductivity in the infinite-layer Nd$_{0.8}$Sr$_{0.2}$NiO$_2$ thin films. The $B_{c\parallel,0°}^{onset}(T)$ is determined at the point where the $R(T)$ curve deviates from the extrapolation of the normal state resistance under applied magnetic field. Above the $B_{c\parallel,0°}^{onset}(T)$, the Nd$_{0.8}$Sr$_{0.2}$NiO$_2$ thin films are in the normal state. The $B_{c\parallel,0°}^{1\%}(T)$ corresponds to the magnetic fields and temperatures below which the resistance $R$ is smaller than 1% of the $R_N$, enclosing the region labeled as zero-resistance state in the phase diagram. The region between the $B_{c\parallel,0°}^{onset}(T)$ and the $B_{c\parallel,0°}^{1\%}(T)$ represents the superconducting transi-tion, and is further separated into three regions labeled as ~isotropy, C$_4$, and C$_4$ + C$_2$, depending on whether the corresponding $R(\varphi)$ curves exhibit nearly isotropy, C$_4$, or C$_4$ + C$_2$ rotational symmetry. More $R(\varphi)$ curves can be found in Supplementary Fig. 14. The representative $R(\varphi)$ curves are shown in Fig. 3b–e, and the arrows approximately mark the temperatures and magnetic fields where the $R(\varphi)$ curves are measured. The $B_{c\parallel,0°}^{50\%}(T)$ and $B_{c\parallel,45°}^{50\%}(T)$ curves determined by the 50% $R_N$ cri-terion are also plotted in the phase diagram.

The phase diagram demonstrates an evolution of the super-conducting states manifesting different rotational symmetries, depending on the external magnetic field. Specifically, in the gray

region labeled as -isotropy ($B < 4$ T), the $B_{c\|,0°}^{50\%}(T)$ and $B_{c\|,45°}^{50\%}(T)$ curves overlap with each other and, consistently, the $R(\varphi)$ curves are nearly isotropic within our measurement resolution, indicating the isotropic superconductivity (Fig. 3e). Under the magnetic field from 4 T to 12 T (the blue region labeled as $C_4$), the $R(\varphi)$ curves exhibit $C_4$ rotational symmetric anisotropy (Fig. 3d), which should be ascribed to the superconductivity of the $Nd_{0.8}Sr_{0.2}NiO_2$ films since it disappears in the normal state (Fig. 2c). Simultaneously, $B_{c\|,0°}^{50\%}(T)$ and $B_{c\|,45°}^{50\%}(T)$ curves split in this region, according with the emergence of $C_4$ anisotropy. When the magnetic field is increased above 12 T (the pink region labeled as $C_4 + C_2$), an additional $C_2$ anisotropy is observed in the $R(\varphi)$ curves as a superimposed modulation on the predominant $C_4$ anisotropy, which breaks the $C_4$ symmetry (Fig. 3b, c). At the same time, the $B_{c\|,0°}^{50\%}(T)$ curve shows an anomalous upturn at the low temperatures above 12 T, deviating from the saturating $B_c$ at the low temperatures expected for a conventional superconducting state. The simultaneous occurrences of the rotational symmetry breaking in the $R(\varphi)$ curves and the enhanced superconducting critical field behavior strongly indicate the emergence of an exotic state.

## Discussion

The superconducting phase diagram may reveal two phase transitions characterized by spontaneous rotational symmetry breakings. The first transition occurs at approximately 4 T indicated by the change from isotropic superconductivity to $C_4$ anisotropy. Considering that the $R(\varphi)$ curves show the symmetry of the in-plane critical field and could reflect the superconducting pairing[37], the first rotational symmetry breaking may suggest a transition from $s$-wave to $d$-wave superconductivity. The transition from $s$-wave to $d$-wave superconductivity is reminiscent of the theoretical phase diagram hosting $s$-, $d$- and $(d+is)$-wave superconductivity for nickelates with varying parameters[14]. Experimentally, the $s$- and $d$-wave mixture has been reported by the previous STM study[15]. The second transition with the rotational symmetry turning from $C_4$ to $C_4 + C_2$ takes place around 12 T. Also, the second transition is accompanied with an anomalous upturn of the in-plane critical field, unveiling the emergence of an electronic state unexplored before. As discussed above, we speculatively ascribe the second transition to the emergence of the charge order in the $Nd_{0.8}Sr_{0.2}NiO_2$ films. It is normally believed that the long-range charge order disfavors the superconductivity[18,22]. In our $Nd_{0.8}Sr_{0.2}NiO_2$ films, with increasing magnetic field, the superconductivity gets suppressed and the stripe charge order fluctuation with short-range correlation emerges, accounting for the relatively small $C_2$ symmetric anisotropy in $R(\varphi)$ curves. The coexistence between the ordered phases with different broken symmetries is relatively rare, and the intertwined orders would give rise to more unexpected quantum phenomena and more complex phase diagram. In our $Nd_{0.8}Sr_{0.2}NiO_2$ thin films, the short-range stripe order coupled to the superconducting condensation may induce a secondary PDW state, in which the superconducting order parameter is oscillatory in space[22,41,42]. Through pairing in the presence of the periodic potential of the charge order, the Cooper pairs gain finite center-of-mass momenta[42], which is also a signature of the Fulde–Ferrell–Larkin–Ovchinnikov (FFLO) phase that features an upturn of the critical magnetic field at the low temperatures[43,44], resembling our observations. These phases were not expected previously in the $Nd_{0.8}Sr_{0.2}NiO_2$ systems. Our findings suggest that the nickelates would be a potential option to explore these exotic states.

Our experimental observations have important implications on current theoretical debates. The isotropic superconductivity requires a primary pairing mechanism that is not expected in optimally hole-doped superconducting cuprates. The successive phase transitions reveal a subtle balance between several competing interactions that are unique for infinite-layer nickelates and also cannot be explained as in cuprates. In the Mott-Kondo scenario, the phase transition may be attributed to the competition between the Kondo coupling and the AFM spin superexchange coupling[14]. The Kondo coupling can produce local spin fluctuations that support isotropic $s$-wave pairing[45], while the AFM coupling favors $d$-wave pairing. In $Nd_{1-x}Sr_xNiO_2$, the superconductivity emerges around the border of the Kondo regime[46]. Therefore, applying magnetic field may suppress the local spin fluctuations, tilt the balance towards AFM correlation, and thus induce a secondary $d$-wave component, which explains the emergent $C_4$ symmetry by either $d$- or $(d+is)$-wave pairing[14]. The charge order also competes with the AFM correlation[19] and the superconductivity. Previous experiments have shown that the charge order phase boundary may even penetrate into the superconducting dome. Further increasing magnetic field may reduce both the superconductivity and AFM correlation, thus promote the charge order, and lead to the observed $C_2$ symmetry. The interplay of the Kondo effect, the AFM correlation, the superconductivity, and the charge order provides a potential playground for novel correlated phenomena, which is well beyond a simple scenario. Any candidate theory should be made in conformity with all these experimental observations.

In summary, we systematically investigated the angular dependence of the transport properties of the infinite-layer nickelate $Nd_{0.8}Sr_{0.2}NiO_2$ thin films with Corbino-disk configuration. The evident polar angular dependent anisotropy strongly indicates the quasi-2D nature of the superconductivity in $Nd_{0.8}Sr_{0.2}NiO_2$ thin films. The azimuthal angular dependence of the magnetoresistance $R(\varphi)$ curves manifests different rotational symmetric structures depending on the external magnetic fields, which are summarized in the $B$ versus $T$ phase diagram. With increasing magnetic field, the $R(\varphi)$ curves exhibit isotropic, $C_4$ symmetric, and then $C_4 + C_2$ symmetric behaviors successively, demonstrating the emergence of the quantum states characterized by spontaneous rotational symmetry breaking. Since the symmetry of the $R(\varphi)$ curves could reflect the symmetry of the superconducting order parameter, the observed $R(\varphi)$ curves evolving from isotropy to $C_4$ anisotropy may suggest an extraordinary superconducting pairing of the $Nd_{0.8}Sr_{0.2}NiO_2$ beyond the familiar cuprate scenario. In the low-temperature and large-magnetic-field regime, we find a striking concurrence of an additional $C_2$ symmetric modulations in the $R(\varphi)$ curves and an anomalous upturn of the temperature-dependent critical magnetic field. We speculate that the $C_2$ feature in the $R(\varphi)$ curves may result from the stripe charge order fluctuations in the $Nd_{0.8}Sr_{0.2}NiO_2$, which can coexist with the superconductivity, and might potentially give rise to a charge order-driven PDW state. Our findings shed new light on the dimensionality and the pairing symmetry of the superconductivity in the infinite-layer nickelate $Nd_{0.8}Sr_{0.2}NiO_2$ thin films, and reveal that the superconducting $Nd_{0.8}Sr_{0.2}NiO_2$ is a promising material platform to study the unconventional superconductivity and the interplay between the various exotic states.

## Methods

### Thin-film synthesis

The $Nd_{0.8}Sr_{0.2}NiO_2$ films with infinite-layer structure are prepared by topochemical reduction of perovskite $Nd_{0.8}Sr_{0.2}NiO_3$ films (thickness about 14.5–16 nm) without capping layer[47]. The precursor $Nd_{0.8}Sr_{0.2}NiO_3$ films are deposited on the $TiO_2$-terminated $SrTiO_3$ (001) substrates by pulsed laser deposition using 248 nm KrF laser. The substrate temperature is controlled at 620 °C with an oxygen pressure of 200 mTorr during the deposition. A laser fluence of 1 J/cm$^2$ was used to ablate the target and the size of laser spot is about 3 mm$^2$. After deposition, the samples were cooled down in the same oxygen pressure at the rate of 10 °C/min. To obtain the infinite-layer phase, the samples were sealed in the quartz tube together with 0.1 g $CaH_2$. The pressure of the tube is about 0.3 mTorr. Then, the tube was heated up to 300 °C in tube furnace, hold for 2 h, and naturally cooled down with the ramp rate of 10 °C/min. The characterizations of the high quality of our samples can be found in the Supplementary Information.

## Devices fabrication

The Corbino-disk electrode is fabricated using the standard photo-lithography technique. After spin coating of PMMA on the $Nd_{0.8}Sr_{0.2}NiO_2$ thin film samples, the annular electrodes were patterned through electron beam lithography in a FEI Helios NanoLab 600i Dual Beam System. Then, the metal electrodes (Ti/Au, 6.5/100 nm) were deposited in a LJUHVE-400 L E-Beam Evaporator. After that, the PMMA layers were removed by the standard lift-off process. Finally, the electrodes are contacted using wire-bonded Al wires for transport measurements.

## Transport measurements

The angle-dependent transport measurements were carried out on a rotator with an accuracy of 0.01° in a 16 T-Physical Property Measurement System (PPMS-EverCool-16, Quantum Design).

## Data availability

All data needed to evaluate the conclusions in the study are present in the paper and/or the Supplementary Information. The data that support the findings of this study are available from the corresponding author upon request.

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

## Acknowledgements
We acknowledge insightful discussions with Yanzhao Liu, Ying Xing, Xiaorong Liu, Mei Wu, Qizhi Li, Yingying Peng, and Huiqian Luo. This work was financially supported by the National Natural Science Foundation of China [Grant No. 11888101 (J.W.)], the National Key Research and Development Program of China [Grant No. 2018YFA0305604 (J.W.), 2022YFA1403103 (Y.L.)], the Innovation Program for Quantum Science and Technology [2021ZD0302403 (J.W.)], Beijing Natural Science Foundation [Z180010 (J.W.)], the National Natural Science Foundation of China [No. 12174442 (Y.L.)], the Strategic Priority Research Program of Chinese Academy of Sciences [Grant No. XDB28000000 (J.W.)], Young Elite Scientists Sponsorship Program by BAST [No. BYESS2023452 (Y.L.)], the Fundamental Research Funds for the Central Universities and the Research Funds of Renmin University of China [Grant No. 22XNKJ20 (Y.L)], the National Natural Science Foundation of China [Grant Nos. 12274061 (L.Q.), 52072059 (L.Q.), and 11774044 (L.Q.)], Science and Technology Department of Sichuan Province [Grant Nos. 2021JDJQ0015 (L.Q.) and 2022ZYD0014 (L.Q.)], the National Natural Science Foundation of China [NSFC Grants No. 11974397 (Y.F.Y)], and the Strategic Priority Research Program of the Chinese Academy of Sciences [Grant No. XDB33010100 (Y.F.Y)].

## Author contributions
J.W. conceived and instructed the research. H.R.J., Y.L., Y.N.L., Z.Y.X., C.C.J. and S.C.Q. fabricated the Corbino-disk devices, performed the transport measurements, and analyzed the data under the guidance of J.W. X.D. and L.Q. synthesized the thin film samples. X.D., M.H.X., L.Q., X.Y.G. and P.G. characterized the thin film samples. Y.F.Y. and G.M.Z. contributed to the theoretical explanations. H.R.J., Y.L., Y.N.L. and J.W. wrote the manuscript with input from all authors.

## Competing interests
The authors declare no competing interests.

## Additional information

[1]International Center for Quantum Materials, School of Physics, Peking University, Beijing 100871, China. [2]Department of Physics and Beijing Key Laboratory of Opto-electronic Functional Materials & Micro-nano Devices, Renmin University of China, Beijing 100872, China. [3]Key Laboratory of Quantum State Construction and Manipulation (Ministry of Education), Renmin University of China, Beijing 100872, China. [4]Department of Physics, The Pennsylvania State University, University Park, PA 16802, USA. [5]School of Physics, University of Electronic Science and Technology of China, Chengdu 610054, China. [6]Electron Microscopy Laboratory, School of Physics, Peking University, Beijing 100871, China. [7]Beijing National Laboratory for Condensed Matter Physics and Institute of Physics, Chinese Academy of Sciences, Beijing 100190, China. [8]School of Physical Sciences, University of Chinese Academy of Sciences, Beijing 100049, China. [9]Songshan Lake Materials Laboratory, Dongguan, Guangdong 523808, China. [10]State Key Laboratory of Low-Dimensional Quantum Physics and Department of Physics, Tsinghua University, Beijing 100084, China. [11]Frontier Science Center for Quantum Information, Beijing 100084, China. [12]Collaborative Innovation Center of Quantum Matter, Beijing 100871, China. [13]CAS Center for Excellence in Topological Quantum Computation, University of Chinese Academy of Sciences, Beijing 100190, China. [14]Beijing Academy of Quantum Information Sciences, Beijing 100193, China. [15]Hefei National Laboratory, Hefei 230088, China. [16]These authors contributed equally: Haoran Ji, Yi Liu, Yanan Li. ✉e-mail: liang.qiao@uestc.edu.cn; jianwangphysics@pku.edu.cn

