## [Peer Review File · Nature Communications]

Rotational symmetry breaking in superconducting nickelate Nd_{0.8} Sr_{0.2} NiO₂ filmsEditorial Note: This manuscript has been previously reviewed at another journal that is not operating a transparent peer review scheme. This document only contains reviewer comments and rebuttal letters for versions considered at *Nature Communications*. Mentions of the other journal have been redacted.

REVIEWERS' COMMENTS

Reviewer #1 (Remarks to the Author):

With regards to the technical questions I posed in the earlier review, the authors have provided a detailed account of the sample preparation, sample quality, and several types of characterization. Several of the results look to have been published separately. While I'm curious to the reason this host of characterization data was omitted from the original report, perhaps the initial omission of the synthetic details was a matter of avoiding overlap between journals.

The data quality remains high, and the figure quality is good. I agree with Reviewer 2's assessment that the study is limited to a singular composition, though I could be convinced that the manuscript is appropriate for publication provided the authors responses.

Reviewer #2 (Remarks to the Author):

I have read through the reviews from both referees along with the rebuttal and revised manuscript. It would appear that the authors have carried out a careful riposte to the various comments, particularly to the comments of Reviewer #1 (though I am sure the reviewer has their own thoughts on this). With regards to my comment about the study being restricted to a single family at a single doping, I appreciate the effort the authors have made to incorporate an additional La-based nickelate film into their study and it is interesting to see that the C4 anisotropy is not seen in the latter. This particular distinction between the Nd- and La-based nickelates was also reported in the Wang paper that I mentioned in my previous report.

The crux of the authors argument (that the C4 anisotropy is related to the

superconductivity and not to the 4f moments of the Nd ions) rests on their observation that the disappearance of the C4 anisotropy coincides with the superconducting transition. While this appears to be true (and was not seen in the Wang study), I do not share the authors' assertion that this 'unambiguously demonstrates that the C4 anisotropy is associated with the unconventional (i.e. d-wave) superconductivity. Unfortunately, the lack of a broader study (of samples with different Sr content demonstrating the coincidence of Tc and the vanishing of the C4 anisotropy for a range of Tc values) weakens this assertion significantly and that is why I am still of the opinion that this article is more suitable for another journal such as npj Quantum Materials. It is a very nice study, but in my opinion, it lacks the definitive evidence to warrant publication in [Redacted].

Response to Reviewers

The reviewers' original comments are shown in blue characters. Our responses are shown in black characters, and the revised contents made to the manuscript are shown in purple characters.

Reviewer #1 (Remarks to the Author):

Comment:

With regards to the technical questions I posed in the earlier review, the authors have provided a detailed account of the sample preparation, sample quality, and several types of characterization. Several of the results look to have been published separately. While I'm curious to the reason this host of characterization data was omitted from the original report, perhaps the initial omission of the synthetic details was a matter of avoiding overlap between journals.

Response: We would like to thank Reviewer #1 again for suggesting us to add the sample characterization information in the last round of review, which have indeed improved our work and may arouse broader interest. We assure that the sample characterization data in our revised manuscript have not been published before.

Comment:

The data quality remains high, and the figure quality is good. I agree with Reviewer 2's assessment that the study is limited to a singular composition, though I could be convinced that the manuscript is appropriate for publication provided the authors responses.

Response: We sincerely appreciate Reviewer #1 for the high evaluation of our work and the recommendation for publication.

With regards to the singular composition in our original report, we have further performed the angular dependent measurements on La-based nickelate in the last round of review. Then, our work becomes more systematic and has involved both Nd-based and La-based nickelate films, instead of singular composition. The angular dependent results of the La-based nickelate film added in our revised manuscript are also appreciated by Reviewer #2, who states that “With regards to my comment about the study being restricted to a single family at a single doping, I appreciate the effort the authors have made to incorporate an additional La-based nickelate film into their study and it is interesting to see that the C₄ anisotropy is not seen in the latter”.

Moreover, the nickelate system, even for singular composition, hosts sufficiently extraordinary phenomena. Many pioneering works are accomplished without changing the rare-earth element nor the Sr content, including: Nature doi: 10.1038/s41586-023-06408-7 based on La₃Ni₂O₇, Nature **615**, 50 (2023) on Nd_{0.8}Sr_{0.2}NiO₂, and Nature Physics **17**, 473 (2021) on Nd_{0.775}Sr_{0.225}NiO₂.

Additionally, we want to mention that our manuscript has been further improved according to the constructive comments from Reviewer #1 about the overshoot of 180° in the two-fold (C₂) anisotropy. To further substantiate the intrinsic C₂ anisotropy, we performed the angular dependent measurements on Nd_{0.8}Sr_{0.2}NiO₂ film using a two-axis rotator instrument, through which the slight misalignment of magnetic field can be eliminated. The obtained azimuthal angular dependent magnetoresistance (R(ϕ)) curve at 16 T shows a nearly perfect C₂ anisotropy superimposed on four-fold (C₄) anisotropy with negligible overshoot issue (Fig. R1b). We have added the corresponding discussions and results in the revised manuscript.

Fig. R1. **a**, Temperature-dependent resistance $R(T)$ of $\text{Nd}_{0.8}\text{Sr}_{0.2}\text{NiO}_2$ film sample S9 measured by the Corbino-disk configuration. **b**, Rectangular plot of azimuthal angle dependent magnetoresistance $R(\varphi)$ curve measured at 16 T and 2 K using a two-axis rotator instrument, where the $\text{Nd}_{0.8}\text{Sr}_{0.2}\text{NiO}_2$ thin film is in the superconducting state. The $R(\varphi)$ curve shows $C_2 + C_4$ rotational symmetry. A nearly perfect C_2 symmetric component can be determined by $R(0^\circ) \approx R(180^\circ) < R(90^\circ) \approx R(270^\circ)$, as indicated by the black dashed lines.

Again, we gratefully thank Reviewer #1 for the recommendation for the publication in [Redacted]. We hope that Reviewer #1 will find our response satisfying and worthy of a timely publication in *Nature Communications*.

Reviewer #2 (Remarks to the Author):

Comment:

I have read through the reviews from both referees along with the rebuttal and revised manuscript. It would appear that the authors have carried out a careful riposte to the various comments, particularly to the comments of Reviewer #1 (though I am sure the reviewer has their own thoughts on this). With regards to my comment about the study

being restricted to a single family at a single doping, I appreciate the effort the authors have made to incorporate an additional La-based nickelate film into their study and it is interesting to see that the C4 anisotropy is not seen in the latter. This particular distinction between the Nd- and La-based nickelates was also reported in the Wang paper that I mentioned in my previous report.

Response: We gratefully thank Reviewer #2 for appreciating our responses and new results. Our angular dependent results are measured with the unique Corbino-disk configuration which naturally has the superiority in measuring the anisotropy of a system, and we are glad to see that the absence of four-fold (C_4) anisotropy in the La-based nickelate is basically consistent with Wang et al.'s results obtained through the four-probe configuration (arXiv: 2205.15355). Moreover, we would like to emphasize that our work exclusively reports the intrinsic two-fold (C_2) anisotropy and the successive rotational symmetry breakings in the nickelate $\text{Nd}_{0.8}\text{Sr}_{0.2}\text{NiO}_2$, representing the uniqueness of our observations and the important implications to the community.

Comment:

The crux of the authors argument (that the C_4 anisotropy is related to the superconductivity and not to the 4f moments of the Nd ions) rests on their observation that the disappearance of the C_4 anisotropy coincides with the superconducting transition. While this appears to be true (and was not seen in the Wang study), I do not share the authors' assertion that this 'unambiguously demonstrates that the C_4 anisotropy is associated with the unconventional (i.e. d-wave) superconductivity. Unfortunately, the lack of a broader study (of samples with different Sr content demonstrating the coincidence of T_c and the vanishing of the C_4 anisotropy for a range of T_c values) weakens this assertion significantly and that is why I am still of the opinion that this article is more suitable for another journal such as npj Quantum Materials. It is a very nice study, but in my opinion, it lacks the definitive evidence to warrant publication in [Redacted].

Response: We gratefully thank Reviewer #2 for the insightful suggestion about a broad study with a range of superconducting transition temperature (T_c). Actually, our results are capable of elucidating the coincidence of T_c and the vanishing of the C_4 anisotropy for a range of T_c values. First, the onset superconducting transition temperature (T_c^{onset}) varies from 10.10 K to 13.25 K with applied magnetic fields in Fig. 2 in the main text, and the C_4 anisotropy indeed disappears when entering the normal state (i.e, when the averaged magnetoresistance R_{avg} reaching the normal state resistance R_N) at different T_c^{onset} values. The corresponding results are summarized in Fig. R2, modified from Fig. 2 in the main text, where the T_c^{onset} varies with applied magnetic fields (B) as labelled in each panel. As the sample is approaching the normal state and losing the superconductivity (R_{avg}/R_N getting close to 100%), the amplitudes of the C_4 anisotropy are diminishing, which shows that the C_4 anisotropy is associated with the superconducting state for a range of T_c under external magnetic fields (Fig. R2). Second, we have measured plenty of samples whose T_c^{onset} ($B = 0$ T) varies from approximately 12.1 K to 15.5 K. To further elucidate our study with a range of T_c , we have shown more azimuthal angular dependent magnetoresistance ($R(\varphi)$) curves showing temperature-dependent C_4 anisotropy for samples with different T_c (Fig. R3). We hope that Reviewer #2 will be satisfied with our systematic study, which demonstrates the coincidence of the superconductivity and the C_4 anisotropy for a range of T_c values. The corresponding discussions and results have also been added in the revised manuscript.

Fig. R2. Four-fold components ΔR_{C4} versus the ratio between the averaged magnetoresistance and the normal state resistance (R_{avg}/R_N) under different in-plane magnetic fields of sample S1. The T_c^{onset} varies with applied magnetic fields and is labelled in each panel. The ΔR_{C4} and R_{avg} values are extracted by trigonometric function fitting. When the R_{avg}/R_N value is getting close to 100%, the temperature is getting close to the onset superconducting transition temperature (T_c^{onset}), and the $Nd_{0.8}Sr_{0.2}NiO_2$ thin film is approaching the normal state and losing the superconductivity. The amplitudes of the C_4 anisotropy are gradually diminishing, showing a tendency that the C_4 anisotropy disappears just when the superconductivity is destroyed at higher temperatures, which strongly indicates that the C_4 anisotropy is associated with the superconducting state for T_c^{onset} ranging from 10.10 K to 13.25 K.

Fig. R3. Temperature-dependent resistance $R(T)$ (a, c, e, g, i) and $R(\phi)$ (b, d, f, h, j) curves obtained from different samples S6 (a and b), S5 (c and d), S4 (e and f), S2 (g and h), S3 (i and j), whose $T_c^{\text{onset}}(B = 0)$ varies from 12.06 K to 15.38 K as labelled in the panels. The C_4 anisotropy in the $R(\phi)$ curves of the superconducting state gradually fades away with increasing temperature, and disappears when approaching the normal state of the samples for all the $\text{Nd}_{0.8}\text{Sr}_{0.2}\text{NiO}_2$ thin films with different $T_c^{\text{onset}}(B = 0)$ values. The reproducible results for a range of T_c^{onset} indicate that the C_4 anisotropy is associated with the superconducting state.

Furthermore, the disappearance of the C_4 anisotropy coinciding with the superconducting transition is consistently observed by another two groups, although they did not use the Corbino-disk configuration nor have they dug into the coincidence between the disappearance of the C_4 anisotropy and the superconducting transition (arXiv: 2205.15355 and arXiv: 2301.07606). Wang et al. concluded that “In the normal state, negligible angular dependence is observed for all samples” (arXiv: 2205.15355). Chow et al. stated that “The normal state azimuthal angular dependent magnetoresistance shows virtually a circle in the polar plot with negligible anisotropy

being observed...observe only the anisotropy in the superconducting states” (arXiv: 2301.07606). Therefore, we believe that our conclusion of superconducting C_4 anisotropy would be widely accepted and appreciated by the community. In addition, we want to mention that, for the in-plane anisotropy in the Nd-based nickelates, Wang et al. only studied four samples with two Sr-contents, i.e., $\text{Nd}_{0.8}\text{Sr}_{0.2}\text{NiO}_2$ and $\text{Nd}_{0.775}\text{Sr}_{0.225}\text{NiO}_2$ (Fig. S5 in arXiv: 2205.15355). According to their Fig. S2c, the T_c ranges merely from approximately 6.5 K to 6.9 K among these samples, representing a very narrow range of T_c values compared with our results (six samples from S1 to S6 with the range of T_c from 12.06 K to 15.38 K).

Moreover, we want to clarify that we have also elaborated to exclude the possibility that the C_4 anisotropy originates from the 4f moment of the Nd ion. As we discussed in the last round of review and in our revised Supplementary Information, the interpretation based on Nd moment cannot explain our observations including: the rotational symmetry breakings with applied magnetic fields (from isotropy to C_4 and from C_4 to C_4+C_2), the emergence of C_2 anisotropy which breaks the C_4 symmetry of the Nd ion distributions, the C_4' anisotropy observed in the normal state above 15 T which has different manifestations and origin compared with the C_4 anisotropy in the superconducting state. Therefore, combined with the coincidence of the disappearance of C_4 anisotropy and the superconducting transition, we attribute the C_4 anisotropy to the superconducting state of the $\text{Nd}_{0.8}\text{Sr}_{0.2}\text{NiO}_2$ films.

In summary, we are very grateful to Reviewer #2's high evaluations and insightful suggestions on our work, which have greatly improved our manuscript. We hope that Reviewer #2 will find our revised manuscript sufficiently convincing and worthy of a timely publication in *Nature Communications*.

REVIEWERS' COMMENTS

Reviewer #1 (Remarks to the Author):

I believe that the changes proposed by the authors in their last round of revisions is sufficient to warrant publication in Nature Communications. I think the journal will suit their study well, both in scope and readership.

Reviewer #2 (Remarks to the Author):

I am satisfied with the revised manuscript and with the addition of new data that helps to support, if not quite confirm, the authors' claim of a link between the emergence of C4 symmetry and superconductivity. I am therefore happy to recommend publication of the manuscript in Nature Communications.

Before publication, however, I do insist that the authors add the correct full citations to all those articles for which there is only an arXiv label - certainly references 16, 18, 19 and 33, which I know have been published in actual journals. There are two others (references 17 and 46) that need to be checked before the paper is published.

Response to Reviewers

The reviewers' original comments are shown in blue characters. Our responses are shown in black characters.

Reviewer #1 (Remarks to the Author):

Comment:

I believe that the changes proposed by the authors in their last round of revisions is sufficient to warrant publication in *Nature Communications*. I think the journal will suit their study well, both in scope and readership.

Response: We express our sincere appreciation to Reviewer #1 for recommending the publication of our manuscript in *Nature Communications*.

Reviewer #2 (Remarks to the Author):

Comment:

I am satisfied with the revised manuscript and with the addition of new data that helps to support, if not quite confirm, the authors' claim of a link between the emergence of C4 symmetry and superconductivity. I am therefore happy to recommend publication of the manuscript in *Nature Communications*.

Before publication, however, I do insist that the authors add the correct full citations to all those articles for which there is only an arXiv label - certainly references 16, 18, 19 and 33, which I know have been published in actual journals. There are two others (references 17 and 46) that need to be checked before the paper is published.

Response: We sincerely appreciate Reviewer #2 for recommending the publication of our manuscript in *Nature Communications*. As suggested by the reviewer, we have checked and updated all the references in our paper.